# Importance-aware Co-teaching for Offline Model-based Optimization

**Ye Yuan[1]\*, Can (Sam) Chen[1,2]\*†, Zixuan Liu[3], Willie Neiswanger[4], Xue Liu[1]**

[1] McGill University, [2] MILA - Quebec AI Institute,
[3] University of Washington, [4] Stanford University
ye.yuan3@mail.mcgill.ca, can.chen@mila.quebec,
zucksliu@cs.washington.edu, neiswanger@cs.stanford.edu,
xueliu@cs.mcgill.ca

## Abstract

Offline model-based optimization aims to find a design that maximizes a property of interest using only an offline dataset, with applications in robot, protein, and molecule design, among others. A prevalent approach is gradient ascent, where a proxy model is trained on the offline dataset and then used to optimize the design. This method suffers from an out-of-distribution issue, where the proxy is not accurate for unseen designs. To mitigate this issue, we explore using a pseudo-labeler to generate valuable data for fine-tuning the proxy. Specifically, we propose *Importance-aware Co-Teaching for Offline Model-based Optimization* (**ICT**). This method maintains three symmetric proxies with their mean ensemble as the final proxy, and comprises two steps. The first step is *pseudo-label-driven co-teaching*. In this step, one proxy is iteratively selected as the pseudo-labeler for designs near the current optimization point, generating pseudo-labeled data. Subsequently, a co-teaching process identifies small-loss samples as valuable data and exchanges them between the other two proxies for fine-tuning, promoting knowledge transfer. This procedure is repeated three times, with a different proxy chosen as the pseudo-labeler each time, ultimately enhancing the ensemble performance. To further improve accuracy of pseudo-labels, we perform a secondary step of *meta-learning-based sample reweighting*, which assigns importance weights to samples in the pseudo-labeled dataset and updates them via meta-learning. ICT achieves state-of-the-art results across multiple design-bench tasks, achieving the best mean rank of 3.1 and median rank of 2, among 15 methods. Our source code can be found here.

## 1 Introduction

A primary goal in many domains is to design or create new objects with desired properties [1]. Examples include the design of robot morphologies [2], protein design, and molecule design [3, 4]. Numerous studies obtain new designs by iteratively querying an unknown objective function that maps a design to its corresponding property score. However, in real-world scenarios, evaluating the objective function can be expensive or risky [3–7]. As a result, it is often more practical to assume access only to an offline dataset of designs and their property scores. This type of problem is referred to as offline model-based optimization (MBO) [1]. The goal of MBO is to find a design that maximizes the unknown objective function using solely the offline dataset.

---

\*Equal contribution with random order.
†Corresponding author.

37th Conference on Neural Information Processing Systems (NeurIPS 2023).

Gradient ascent is a common approach to address the offline MBO problem. For example, as illustrated in Figure 2 (a), the offline dataset may consist of three robot size and robot speed pairs $p_{1,2,3}$. A simple DNN model, referred to as the *vanilla proxy* and represented as $f_{\boldsymbol{\theta}}(\cdot)$, is trained to fit the offline dataset as an approximation to the unknown objective function. Gradient ascent is subsequently applied to existing designs with respect to the vanilla proxy $f_{\boldsymbol{\theta}}(\cdot)$, aiming to generate a new design with a higher score. However, the gradient ascent method suffers from an out-of-distribution issue, where the vanilla proxy cannot accurately estimate data outside of the training distribution, leading to a significant gap between the vanilla proxy and the ground-truth function, as shown in Figure 2 (a). As a consequence, the scores of new designs obtained via gradient ascent can be erroneously high [8, 9].

To mitigate the out-of-distribution issue, recent studies have suggested applying regularization techniques to either the proxy itself [8–10] or the design under consideration [11, 12]. These methods improve the proxy's robustness and generalization ability. However, a yet unexplored approach in this domain is using a pseudo-labeler to assign pseudo-labels to designs near the current point. Fine-tuning the proxy on this pseudo-labeled dataset can lead to improvement, provided that we can identify the valuable portion of the pseudo-labeled dataset.

Inspired by this, we propose ***I**mportance-aware **C**o-**T**eaching for Offline Model-based Optimization* (**ICT**). This approach maintains three symmetric proxies, and their mean ensemble acts as the final proxy. ICT consists of two main steps with the **first step** being *pseudo-label-driven co-teaching* as illustrated in Figure 1. During this step, one proxy is iteratively selected as the pseudo-labeler, followed by a co-teaching process [13] that facilitates the exchange of valuable data between the other two proxies for fine-tuning. As depicted in Figure 1, there are three symmetric proxies, $f_{\boldsymbol{\theta}_1}(\cdot)$, $f_{\boldsymbol{\theta}_2}(\cdot)$, and $f_{\boldsymbol{\theta}_3}(\cdot)$. The entire learning cycle (the larger triangle) can be divided into three symmetric parts (sub-triangles), with one proxy chosen to be the

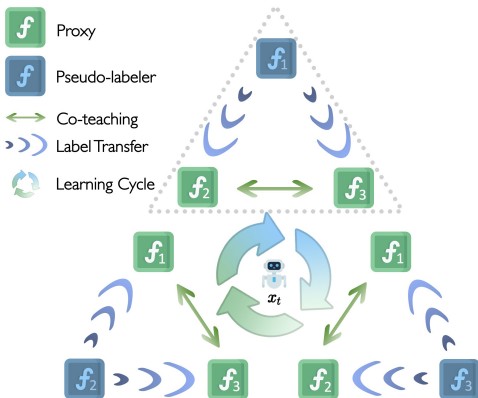

Figure 1: Pseudo-label-driven co-teaching.

pseudo-labeler in turn. Taking the top triangle as an example, we select $f_{\boldsymbol{\theta}_1}(\cdot)$ as the pseudo-labeler to generate pseudo labels for a set of points in the neighborhood of the current optimization point $\boldsymbol{x}_t$. The other two proxies, $f_{\boldsymbol{\theta}_2}(\cdot)$ and $f_{\boldsymbol{\theta}_3}(\cdot)$, then receive the pseudo-labeled dataset. They compute the sample loss for each entry in the dataset and exchange small-loss samples between them for fine-tuning. This co-teaching process encourages knowledge transfer between the two proxies, as small losses are typically indicative of valuable knowledge. The symmetric nature of the three proxies allows the above process to repeat three times, with each proxy—$f_{\boldsymbol{\theta}_1}(\cdot)$, $f_{\boldsymbol{\theta}_2}(\cdot)$, and $f_{\boldsymbol{\theta}_3}(\cdot)$—taking turns as the pseudo-label generator. This learning cycle promotes the sharing of valuable knowledge among the three symmetric proxies, allowing them to collaboratively improve the ensemble performance in handling out-of-distribution designs.

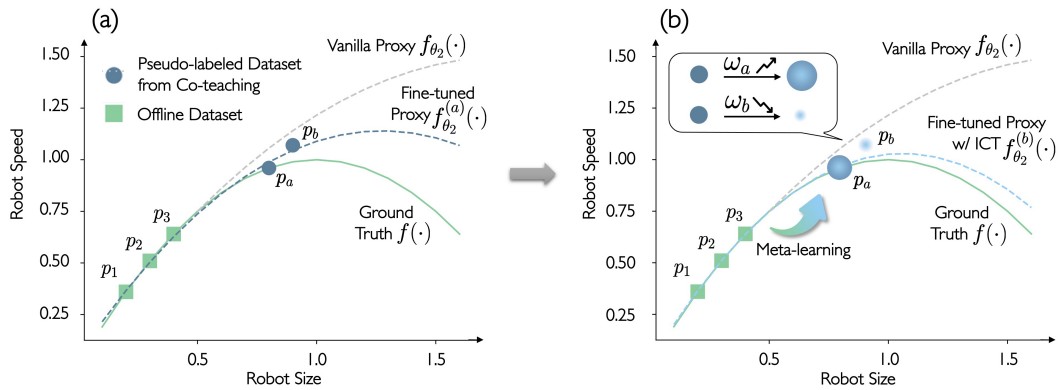

Figure 2: Meta-learning-based sample reweighting.

Despite the efforts made in the first step, small-loss data may still contain inaccurate labels. During the first step, small-loss data ($p_a$ and $p_b$) from the pseudo-labeled dataset produced by $f_{\boldsymbol{\theta}_1}(\cdot)$ are identified based on the predictions of proxy $f_{\boldsymbol{\theta}_3}(\cdot)$ and fed to proxy $f_{\boldsymbol{\theta}_2}(\cdot)$. However, as shown in Figure 2 (a), the less accurate point $p_b$ deviates noticeably from the ground-truth, causing the fine-tuned proxy $f_{\boldsymbol{\theta}_2}(\cdot)$ to diverge from the ground-truth function. To address this, we introduce the **second step** of ICT, *meta-learning-based sample reweighting*, which aims to assign higher weights to more accurate points like $p_a$ and lower weights to less accurate ones like $p_b$. To accomplish this, we assign an importance weight for every sample yielded by the first step ($\boldsymbol{\omega}_a$ for $p_a$ and $\boldsymbol{\omega}_b$ for $p_b$) and propose a meta-learning framework to update these sample weights ($\boldsymbol{\omega}_a$ and $\boldsymbol{\omega}_b$) automatically by leveraging the supervision signals from the offline dataset $p_{1,2,3}$. Specifically, the proxy fine-tuned on the weighted small-loss data ($p_a$ and $p_b$) is expected to perform well on the offline dataset, provided the weights are accurate, i.e., large $\boldsymbol{\omega}_a$ and small $\boldsymbol{\omega}_b$. We can optimize the sample weights by minimizing the loss on the offline dataset as a function of the sample weights. As illustrated in Figure 2 (b), the weight of $p_a$ is optimized to be high, while the weight of $p_b$ is optimized to be low. Consequently, the proxy $f_{\boldsymbol{\theta}_2}^{(b)}(\cdot)$ fine-tuned on the weighted samples in Figure 2 (b) is brought closer to the ground-truth objective function $f(\cdot)$, compared to the case where the fine-tuned proxy $f_{\boldsymbol{\theta}_2}^{(a)}(\cdot)$ is far from $f(\cdot)$ in Figure 2 (a). Through extensive experiments across various tasks [1], ICT proves effective at mitigating out-of-distribution issues, delivering state-of-the-art results.

In summary, our paper presents three main contributions:

- We introduce ***I**mportance-aware **C**o-**T**eaching* (**ICT**) for offline MBO. ICT consists of two steps. In the *pseudo-label-driven co-teaching* step, a proxy is iteratively chosen as the pseudo-labeler, initiating a co-teaching process that facilitates knowledge exchange between the other two proxies.

- The second step, *meta-learning-based sample reweighting*, is introduced to alleviate potential inaccuracies in pseudo-labels. In this step, pseudo-labeled samples are assigned importance weights, which are then optimized through meta-learning.

- Extensive experiments demonstrate ICT's effectiveness in addressing out-of-distribution issues, yielding state-of-the-art results in multiple MBO tasks. Specifically, ICT secures the best mean rank of 3.1 and median rank of 2, among 15 methods.

## 2 Preliminaries

Offline model-based optimization (MBO) targets a variety of optimization problems with the goal of maximizing an unknown objective function using an offline dataset. Consider the design space $\mathcal{X} = \mathbb{R}^d$, where $d$ represents the design dimension. Formally, the offline MBO can be expressed as:

$$\boldsymbol{x}^* = \arg\max_{\boldsymbol{x} \in \mathcal{X}} f(\boldsymbol{x}), \tag{1}$$

where $f(\cdot)$ denotes the unknown objective function, and $\boldsymbol{x} \in \mathcal{X}$ denotes a candidate design. In this scenario, an offline dataset $\mathcal{D} = \{(\boldsymbol{x}_i, y_i)\}_{i=1}^N$ is available, where $\boldsymbol{x}_i$ represents a specific design, such as robot size, and $y_i$ represents the corresponding score, like robot speed. In addition to robot design, similar problems also include protein and molecule design.

A common strategy for tackling offline MBO involves approximating the unknown objective function $f(\cdot)$ using a proxy function, typically represented by a deep neural network (DNN) $f_{\boldsymbol{\theta}}(\cdot)$, which is trained on the offline dataset:

$$\boldsymbol{\theta}^* = \arg\min_{\boldsymbol{\theta}} \frac{1}{N} \sum_{i=1}^N \left(f_{\boldsymbol{\theta}}(\boldsymbol{x}_i) - y_i\right)^2. \tag{2}$$

With the trained proxy, design optimization is performed using gradient ascent steps:

$$\boldsymbol{x}_t = \boldsymbol{x}_{t-1} + \eta \nabla_{\boldsymbol{x}} f_{\boldsymbol{\theta}}(\boldsymbol{x})\Big|_{\boldsymbol{x} = \boldsymbol{x}_t}, \quad \text{for } t \in [1, T]. \tag{3}$$

Here, $T$ denotes the number of steps, and $\eta$ signifies the learning rate. The optimal design $\boldsymbol{x}^*$ is acquired as $\boldsymbol{x}_T$. This gradient ascent approach is limited by an *out-of-distribution issue*, as the proxy $f_{\boldsymbol{\theta}}(\boldsymbol{x})$ may not accurately predict scores for unseen designs, leading to suboptimal solutions.

# 3   Method

In this section, we introduce ***Importance-aware Co-Teaching*** (**ICT**), which consists of two steps. We maintain three symmetric proxies and compute the mean ensemble as the final proxy. In Sec 3.1, we describe the first step, *pseudo-label-driven co-teaching*. This step involves iteratively selecting one proxy as the pseudo-label generator and implementing a co-teaching process to facilitate the exchange of valuable data between the remaining two proxies. Nevertheless, the samples exchanged during co-teaching might still contain inaccurate labels, which necessitates the second step *meta-learning-based sample reweighting* in Sec 3.2. During this step, each sample from the previous step is assigned an importance weight and updated via meta-learning. Intuitively, the ICT process can be likened to an enhanced paper peer review procedure between three researchers preparing for submission. Each researcher, acting as an author, presents his/her paper to the other two. These two serve as reviewers and co-teach each other important points to better comprehend the paper, ultimately providing their feedback to the author. A detailed depiction of the entire algorithm can be found in Algorithm 1.

## 3.1   Pseudo-label-driven Co-teaching

Vanilla gradient ascent, as expressed in Eq. (3), is prone to out-of-distribution issues in offline model-based optimization. One potential yet unexplored solution is using a pseudo-labeler to provide pseudo-labels to designs around the optimization point. By fine-tuning the proxy using the valuable portion of the pseudo-labeled dataset, we can enhance the proxy's performance. To achieve this, we maintain three proxies simultaneously, computing their mean ensemble as the final proxy, and iteratively select one proxy to generate pseudo-labeled data. The other two proxies exchange knowledge estimated to have high value, by sharing small-loss data. Due to the symmetric nature of the three proxies, this process can be repeated three times for sharing valuable knowledge further.

**Pseudo-label.** We initially train three proxies $f_{\boldsymbol{\theta}_1}(\cdot)$, $f_{\boldsymbol{\theta}_2}(\cdot)$, and $f_{\boldsymbol{\theta}_3}(\cdot)$ on the whole offline dataset using Eq. (2) with different initializations, and conduct gradient ascent with their mean ensemble,

$$\boldsymbol{x}_t = \boldsymbol{x}_{t-1} + \eta \nabla_{\boldsymbol{x}} \frac{1}{3}(f_1(\boldsymbol{x}_{t-1}) + f_2(\boldsymbol{x}_{t-1}) + f_3(\boldsymbol{x}_{t-1})), \tag{4}$$

where $\eta$ is the gradient ascent learning rate. Given the current optimization point $\boldsymbol{x}_t$, we sample $M$ points $\boldsymbol{x}_{t,1}, \boldsymbol{x}_{t,2}, \ldots, \boldsymbol{x}_{t,M}$ around $\boldsymbol{x}_t$ as $\boldsymbol{x}_{t,m} = \boldsymbol{x}_t + \gamma\epsilon$, where $\gamma$ is the noise coefficient and $\epsilon$ is drawn from the standard Gaussian distribution. An alternative way is to directly sample the $M$ points around the offline dataset, rather than the current optimization point. We detail this option in Appendix A.1. We iteratively choose one proxy, for example $f_{\boldsymbol{\theta}_1}(\cdot)$, to label these points, creating a pseudo-labeled dataset $\mathcal{D}_1 = \{(\boldsymbol{x}_{t,j}, f_{\boldsymbol{\theta}_1}(\boldsymbol{x}_{t,j}))\}_{j=1}^{M}$. Lines 5 to 6 of Algorithm 1 detail the implementation of this segment.

**Co-teaching.** For each sample in the pseudo-labeled dataset $\mathcal{D}_1$, we compute the sample loss for $f_{\boldsymbol{\theta}_2}(\cdot)$ and $f_{\boldsymbol{\theta}_3}(\cdot)$. Specifically, the losses are calculated as $\mathcal{L}_{2,i} = (f_{\boldsymbol{\theta}_2}(\boldsymbol{x}_{t,i}) - f_{\boldsymbol{\theta}_1}(\boldsymbol{x}_{t,i}))^2$ and $\mathcal{L}_{3,i} = (f_{\boldsymbol{\theta}_3}(\boldsymbol{x}_{t,i}) - f_{\boldsymbol{\theta}_1}(\boldsymbol{x}_{t,i}))^2$, respectively. Small-loss samples typically contain valuable knowledge, making them ideal for enhancing proxy robustness [13]. Proxies $f_{\boldsymbol{\theta}_2}(\cdot)$ and $f_{\boldsymbol{\theta}_3}(\cdot)$ then exchange the top $K$ small-loss samples as valuable data to teach each other where $K$ is a hyperparameter. The co-teaching process enables the exchange of valuable knowledge between proxies $f_{\boldsymbol{\theta}_2}(\cdot)$ and $f_{\boldsymbol{\theta}_3}(\cdot)$. This part is implemented as described in Lines 7 to 8 of Algorithm 1. The symmetric design of the three proxies, $f_{\boldsymbol{\theta}_1}(\cdot)$, $f_{\boldsymbol{\theta}_2}(\cdot)$, and $f_{\boldsymbol{\theta}_3}(\cdot)$, enables the entire process to be iterated three times with one proxy chosen as the pseudo-labeler every time.

## 3.2   Meta-learning-based Sample Reweighting

While the previous step effectively selects samples for fine-tuning, these samples may still contain inaccuracies. To mitigate this, we introduce a *meta-learning-based sample reweighting* step. In this step, each sample obtained from the prior step is assigned an importance weight, which is then updated using a meta-learning framework. Without loss of generality, we use $f_{\boldsymbol{\theta}}(\cdot)$ to represent any of $f_{\boldsymbol{\theta}_1}(\cdot)$, $f_{\boldsymbol{\theta}_2}(\cdot)$ and $f_{\boldsymbol{\theta}_3}(\cdot)$ as this step applies identically to all three proxies. The top $K$ small-loss samples selected from the previous step for fine-tuning $f_{\boldsymbol{\theta}}(\cdot)$ are denoted as $\mathcal{D}_s = \{(\boldsymbol{x}_i^s, \bar{y}_i^s)\}_{i=1}^{K}$.

**Sample Reweighting.** We assign an importance weight $\boldsymbol{\omega}_i$ to the $i^{th}$ selected sample and initialize these importance weights to ones. We expect smaller importance weights for less accurate samples

and larger importance weights for more accurate samples to improve proxy fine-tuning. With these weights, we can optimize the proxy parameters as follows:

$$\boldsymbol{\theta}^*(\boldsymbol{\omega}) = \arg\min_{\boldsymbol{\theta}} \frac{1}{K} \sum_{i=1}^{K} \boldsymbol{\omega_i}(f_{\boldsymbol{\theta}}(\boldsymbol{x}_i^s) - \bar{y}_i^s)^2. \tag{5}$$

Since we only want to perform fine-tuning based on $\mathcal{D}_s$, we can adopt one step of gradient descent:

$$\boldsymbol{\theta}^*(\boldsymbol{\omega}) = \boldsymbol{\theta} - \frac{\alpha}{K} \sum_{i=1}^{K} \boldsymbol{\omega}_i \frac{\partial (f_{\boldsymbol{\theta}}(\boldsymbol{x}_i^s) - \bar{y}_i^s)^2}{\partial \boldsymbol{\theta}^\top}, \tag{6}$$

where $\alpha$ is the learning rate for fine-tuning. This part is presented in Line 10 in Algorithm 1.

**Meta-learning.** The challenge now is finding a group of proper weights $\boldsymbol{\omega}$. We achieve this by leveraging the supervision signals from the offline dataset, which are generally accurate. If the sample weights are accurate, the proxy fine-tuned on the weighted samples is expected to perform well on the offline dataset. This is because the weighted samples aim to reflect the underlying ground-truth function that the offline dataset already captures, and both sets of data share common patterns. We can optimize the sample weights by minimizing the loss of the offline dataset in a meta-learning framework. The loss on the offline dataset can be written as:

$$\mathcal{L}(\boldsymbol{\theta}^*(\boldsymbol{\omega})) = \arg\min_{\boldsymbol{\omega}} \frac{1}{N} \sum_{i=1}^{N} (f_{\boldsymbol{\theta}^*(\boldsymbol{\omega})}(\boldsymbol{x}_i) - y_i)^2. \tag{7}$$

The sample weight $\boldsymbol{\omega}_i$ for the $i^{th}$ sample can be updated by gradient descent:

$$\begin{aligned} \boldsymbol{\omega}_i^{'} &= \boldsymbol{\omega}_i - \beta \frac{\partial \mathcal{L}(\boldsymbol{\theta}^*(\boldsymbol{\omega}))}{\partial \boldsymbol{\theta}} \frac{\partial \boldsymbol{\theta}^*(\boldsymbol{\omega})}{\partial \boldsymbol{\omega}_i} \\ &= \boldsymbol{\omega}_i + \frac{\alpha\beta}{K} \frac{\partial \mathcal{L}(\boldsymbol{\theta}^*(\boldsymbol{\omega}))}{\partial \boldsymbol{\theta}} \frac{\partial (f_{\boldsymbol{\theta}}(\boldsymbol{x}_i^s) - \bar{y}_i^s)^2}{\partial \boldsymbol{\theta}^\top}, \end{aligned} \tag{8}$$

where $\beta$ is the learning rate for the meta-learning framework. From Eq. (8), it is worth mentioning that $\frac{\partial \mathcal{L}(\boldsymbol{\theta}^*(\boldsymbol{\omega}))}{\partial \boldsymbol{\theta}} \frac{\partial (f_{\boldsymbol{\theta}}(\boldsymbol{x}_i^s) - \bar{y}_i^s)^2}{\partial \boldsymbol{\theta}^\top}$ represents the similarity between the gradient of the offline dataset and the gradient of the $i^{th}$ sample. This implies that a sample with a gradient similar to the offline dataset will receive a higher weight and vice versa, revealing the inner mechanism of this framework. By applying the updated sample weights to Eq. (6) for fine-tuning, we improve the proxy's performance. This process is iteratively applied to each proxy, yielding a stronger ensemble. Lines 11 to 13 of Algorithm 1 showcase the execution of this part.

## 4 Experimental Results

### 4.1 Dataset and Evaluation

**Dataset and Tasks.** In this study, we conduct experiments on four continuous tasks and three discrete tasks. The continuous tasks include: (a) Superconductor (SuperC)[5], where the objective is to develop a superconductor with 86 continuous components to maximize critical temperature, using $17,010$ designs; (b) Ant Morphology (Ant)[1, 14], where the aim is to design a quadrupedal ant with 60 continuous components to improve crawling speed, based on $10,004$ designs; (c) D'Kitty Morphology (D'Kitty)[1, 15], where the focus is on shaping a quadrupedal D'Kitty with 56 continuous components to enhance crawling speed, using $10,004$ designs; (d) Hopper Controller (Hopper)[1], where the aim is to identify a neural network policy with $5,126$ weights to optimize return, using $3,200$ designs. Additionally, our discrete tasks include: (e) TF Bind 8 (TF8)[6], where the goal is to discover an 8-unit DNA sequence that maximizes binding activity score, utilizing $32,898$ designs; (f) TF Bind 10 (TF10)[6], where the aim is to find a 10-unit DNA sequence that optimizes binding activity score, using $50,000$ designs; (g) NAS [16], where the objective is to find the optimal neural network architecture to enhance test accuracy on the CIFAR-10 [17] dataset, using $1,771$ designs.

**Evaluation and Metrics.** In accordance with the evaluation protocol used in [1, 11], we identify the top 128 designs from the offline dataset for each approach and report the $100^{th}$ percentile normalized

---

**Algorithm 1** Importance-aware Co-teaching

---

**Input:** Proxies $f_{\boldsymbol{\theta}_1}, f_{\boldsymbol{\theta}_2}, f_{\boldsymbol{\theta}_3}$ with parameters $\boldsymbol{\theta}_1, \boldsymbol{\theta}_2, \boldsymbol{\theta}_3$; Offline dataset $\mathcal{D} = \{(\boldsymbol{x}_i, y_i)\}_{i=1}^N$.
**Output:** High-scoring design $\boldsymbol{x}^*$.

1  /* *Start the main steps of ICT* */
2  $\boldsymbol{x}_0 \longleftarrow$ the design with highest score in $\mathcal{D}$.
3  **for** t = 1, 2, ..., $T$ **do**
4      /* *Pseudo-label-driven Co-teaching* */
5      Sample $M$ points $\boldsymbol{x}_{t,1}, \boldsymbol{x}_{t,2}, \ldots, \boldsymbol{x}_{t,M}$ around $\boldsymbol{x}_t$.
6      Choose a proxy, such as $f_{\boldsymbol{\theta}_1}(\cdot)$, to pseudo-label: $\mathcal{D}_1 \longleftarrow \{(\boldsymbol{x}_{t,j}, f_{\boldsymbol{\theta}_1}(\boldsymbol{x}_{t,j}))\}_{j=1}^M$.
7      Identify the top $K$ small-loss samples in $\mathcal{D}_1$ using $f_{\boldsymbol{\theta}_3}(\cdot)$ and feed them to $f_{\boldsymbol{\theta}_2}(\cdot)$.
8      Identify the top $K$ small-loss samples in $\mathcal{D}_1$ using $f_{\boldsymbol{\theta}_2}(\cdot)$ and feed them to $f_{\boldsymbol{\theta}_3}(\cdot)$.
9      /* *Meta-learning-based Sample Reweighting* */
10     Initialize the sample weights $\boldsymbol{\omega}^{(2)}, \boldsymbol{\omega}^{(3)}$ as ones and perform fine-tuning with Eq. 6).
11     Update $\boldsymbol{\omega}^{(2)}$ for $f_{\boldsymbol{\theta}_2}(\cdot)$ and $\boldsymbol{\omega}^{(3)}$ for $f_{\boldsymbol{\theta}_3}(\cdot)$ with Eq. (8).
12     Update $\boldsymbol{\theta}_2$ using updated weights $\boldsymbol{\omega}^{(2)}$ and $\boldsymbol{\theta}_3$ for $\boldsymbol{\omega}^{(3)}$ with Eq. (6).
13     Repeat Line 6 - 12 with $f_{\boldsymbol{\theta}_2}(\cdot), f_{\boldsymbol{\theta}_3}(\cdot)$ iteratively as pseudo-labeler in Line 6.
14 /* *Finished ICT, start to optimize designs* */
15 **for** t = 1, 2, ..., $T$ **do**
16     Optimize $\boldsymbol{x}_t$ with Eq. (4) based on the mean ensemble of fine-tuned $f_{\boldsymbol{\theta}_1}, f_{\boldsymbol{\theta}_2}, f_{\boldsymbol{\theta}_3}$.
17 **return** $\boldsymbol{x}^* \longleftarrow \boldsymbol{x}_T$

---

ground-truth score. This score is computed as $y_n = \frac{y - y_{min}}{y_{max} - y_{min}}$, where $y_{min}$ and $y_{max}$ represent the minimum and maximum scores within the entire unobserved dataset, respectively. The $50^{th}$ percentile (median) normalized ground-truth scores are included in Appendix A.2. For a better comparison, we report the best design in the offline dataset, denoted as $\mathcal{D}(\textbf{best})$. We also provide mean and median rankings across all seven tasks for a broad performance assessment.

## 4.2 Comparison Methods

We compare our approach with two categories of baselines: (1) those that use generative models for sampling purposes, and (2) those that apply gradient updates derived from existing designs. The generative model-based methods learn and sample from the distribution of high-scoring designs, including: **(i)** MIN [18], which maps scores to designs and searches this map for optimal designs; **(ii)** CbAS [19], which uses a VAE model to adapt the design distribution towards high-scoring areas; **(iii)** Auto.CbAS [20], which employs importance sampling to retrain a regression model based on CbAS.

The second category encompasses: **(i)** Grad: carries out a basic gradient ascent on existing designs to generate new ones; **(ii)** Grad. Min: optimizes the lowest prediction from an ensemble of learned objective functions; **(iii)** Grad. Mean: optimizes the ensemble's mean prediction; **(iv)** ROMA [8]: applies smoothness regularization on the DNN; **(v)** COMs [9]: uses regularization to assign lower scores to designs obtained through gradient ascent; **(vi)** NEMO [10]: constrains the gap between the proxy and the ground-truth function via normalized maximum likelihood before performing gradient ascent; **(vii)** BDI [11] uses forward and backward mappings to distill knowledge from the offline dataset to the design; **(viii)** IOM [21]: enforces representation invariance between the training dataset and the optimized designs.

We also compare with traditional methods in [1]: **(i)** CMA-ES [22]: gradually adjusts the distribution towards the optimal design by modifying the covariance matrix. **(ii)** BO-qEI [23]: executes Bayesian Optimization to maximize the proxy, suggests designs through the quasi-Expected-Improvement acquisition function, and labels the designs using the proxy function. **(iii)** REINFORCE [24]: optimizes the distribution over the input space using the learned proxy.

## 4.3 Training Details

We adopt the training settings from [1] for all comparison methods unless otherwise specified. We use a 3-layer MLP (MultiLayer Perceptron) with ReLU activation for all gradient updating methods, and set the hidden size to 2048. Additional hyperparameter details are elaborated in Appendix A.3.

One of the top 128 designs from the offline dataset is iteratively selected as the starting point, as outlined in Line 2 of Algorithm 1. We reference results from [1] for non-gradient-ascent methods such as BO-qEI, CMA-ES, REINFORCE, CbAS, and Auto.CbAS. For gradient-based methods, we run each setting over 8 trials and report the mean and standard error. All experiments are run on a single NVIDIA GeForce RTX 3090 GPU.

## 4.4 Results and Analysis

**Performance in Continuous Tasks.** Table 1 presents the results across different continuous domains. In all four continuous tasks, our ICT method achieves the top performance. Notably, it surpasses the basic gradient ascent, Grad, demonstrating its ability to mitigate the out-of-distribution issue. The superior performance of Grad.mean over Grad can be attributed to the ensemble model's robustness in making predictions [25]. Furthermore, ICT generally outperforms ensemble methods and other gradient-based techniques such as COMs and ROMA, demonstrating the effectiveness of our strategy. Generative model-based methods, such as CbAS and MINs, however, struggle with the high-dimensional task Hopper Controller. Interestingly, ICT necessitates only three standard proxies and avoids the need for training a generative model, which can often be a challenging task. These results indicate that ICT is a simple yet potent baseline for offline MBO.

Table 1: Experimental results on continuous tasks for comparison.

| Method | Superconductor | Ant Morphology | D'Kitty Morphology | Hopper Controller |
|---|---|---|---|---|
| $\mathcal{D}(\textbf{best})$ | 0.399 | 0.565 | 0.884 | 1.0 |
| BO-qEI | $0.402 \pm 0.034$ | $0.819 \pm 0.000$ | $0.896 \pm 0.000$ | $0.550 \pm 0.018$ |
| CMA-ES | $0.465 \pm 0.024$ | $\textbf{1.214} \pm \textbf{0.732}$ | $0.724 \pm 0.001$ | $0.604 \pm 0.215$ |
| REINFORCE | $0.481 \pm 0.013$ | $0.266 \pm 0.032$ | $0.562 \pm 0.196$ | $-0.020 \pm 0.067$ |
| CbAS | $\textbf{0.503} \pm \textbf{0.069}$ | $0.876 \pm 0.031$ | $0.892 \pm 0.008$ | $0.141 \pm 0.012$ |
| Auto.CbAS | $0.421 \pm 0.045$ | $0.882 \pm 0.045$ | $0.906 \pm 0.006$ | $0.137 \pm 0.005$ |
| MIN | $0.499 \pm 0.017$ | $0.445 \pm 0.080$ | $0.892 \pm 0.011$ | $0.424 \pm 0.166$ |
| Grad | $0.483 \pm 0.025$ | $0.920 \pm 0.044$ | $\textbf{0.954} \pm \textbf{0.010}$ | $\textbf{1.791} \pm \textbf{0.182}$ |
| Mean | $0.497 \pm 0.011$ | $0.943 \pm 0.012$ | $\textbf{0.961} \pm \textbf{0.012}$ | $\textbf{1.815} \pm \textbf{0.111}$ |
| Min | $\textbf{0.505} \pm \textbf{0.017}$ | $0.910 \pm 0.038$ | $0.936 \pm 0.006$ | $0.543 \pm 0.010$ |
| COMs | $0.472 \pm 0.024$ | $0.828 \pm 0.034$ | $0.913 \pm 0.023$ | $0.658 \pm 0.217$ |
| ROMA | $\textbf{0.510} \pm \textbf{0.015}$ | $0.917 \pm 0.030$ | $0.927 \pm 0.013$ | $1.740 \pm 0.188$ |
| NEMO | $0.502 \pm 0.002$ | $0.952 \pm 0.002$ | $\textbf{0.950} \pm \textbf{0.001}$ | $0.483 \pm 0.005$ |
| BDI | $\textbf{0.513} \pm \textbf{0.000}$ | $0.906 \pm 0.000$ | $0.919 \pm 0.000$ | $\textbf{1.993} \pm \textbf{0.000}$ |
| IOM | $\textbf{0.520} \pm \textbf{0.018}$ | $0.918 \pm 0.031$ | $0.945 \pm 0.012$ | $1.176 \pm 0.452$ |
| $\textbf{ICT}_{(\text{ours})}$ | $\textbf{0.503} \pm \textbf{0.017}$ | $\textbf{0.961} \pm \textbf{0.007}$ | $\textbf{0.968} \pm \textbf{0.020}$ | $\textbf{2.104} \pm \textbf{0.357}$ |

Table 2: Experimental results on discrete tasks, and ranking on all tasks for comparison.

| Method | TF Bind 8 | TF Bind 10 | NAS | Rank Mean | Rank Median |
|---|---|---|---|---|---|
| $\mathcal{D}(\textbf{best})$ | 0.439 | 0.467 | 0.436 | | |
| BO-qEI | $0.798 \pm 0.083$ | $0.652 \pm 0.038$ | $\textbf{1.079} \pm \textbf{0.059}$ | 9.9/15 | 11/15 |
| CMA-ES | $\textbf{0.953} \pm \textbf{0.022}$ | $0.670 \pm 0.023$ | $0.985 \pm 0.079$ | 6.1/15 | 3/15 |
| REINFORCE | $\textbf{0.948} \pm \textbf{0.028}$ | $0.663 \pm 0.034$ | $-1.895 \pm 0.000$ | 11.3/15 | 15/15 |
| CbAS | $0.927 \pm 0.051$ | $0.651 \pm 0.060$ | $0.683 \pm 0.079$ | 9.1/15 | 9/15 |
| Auto.CbAS | $0.910 \pm 0.044$ | $0.630 \pm 0.045$ | $0.506 \pm 0.074$ | 11.6/15 | 12/15 |
| MIN | $0.905 \pm 0.052$ | $0.616 \pm 0.021$ | $0.717 \pm 0.046$ | 11.0/15 | 12/15 |
| Grad | $0.906 \pm 0.024$ | $0.635 \pm 0.022$ | $0.598 \pm 0.034$ | 7.7/15 | 9/15 |
| Mean | $0.899 \pm 0.025$ | $0.652 \pm 0.020$ | $0.666 \pm 0.062$ | 6.6/15 | 6/15 |
| Min | $0.939 \pm 0.013$ | $0.638 \pm 0.029$ | $0.705 \pm 0.011$ | 7.3/15 | 8/15 |
| COMs | $0.452 \pm 0.040$ | $0.624 \pm 0.008$ | $0.810 \pm 0.029$ | 10.3/15 | 12/15 |
| ROMA | $0.924 \pm 0.040$ | $0.666 \pm 0.035$ | $0.941 \pm 0.020$ | 5.1/15 | 5/15 |
| NEMO | $0.941 \pm 0.000$ | $\textbf{0.705} \pm \textbf{0.000}$ | $0.734 \pm 0.015$ | 5.0/15 | 4/15 |
| BDI | $0.870 \pm 0.000$ | $0.605 \pm 0.000$ | $0.722 \pm 0.000$ | 7.9/15 | 8/15 |
| IOM | $0.878 \pm 0.069$ | $0.648 \pm 0.023$ | $0.274 \pm 0.021$ | 7.6/15 | 6/15 |
| $\textbf{ICT}_{(\text{ours})}$ | $\textbf{0.958} \pm \textbf{0.008}$ | $\textbf{0.691} \pm \textbf{0.023}$ | $0.667 \pm 0.091$ | $\textbf{3.1/15}$ | $\textbf{2/15}$ |

**Performance in Discrete Tasks.** Table 2 showcases the outcomes across various discrete domains. ICT attains top performances in two out of the three tasks, TF Bind 8 and TF Bind 10. These results suggest that ICT is a powerful method in the discrete domain. However, in NAS, the performance of ICT is not as strong, which can be attributed to two factors. Firstly, the neural network design in NAS,

represented by a 64-length sequence of 5-categorical one-hot vectors, has a higher dimensionality than TF Bind 8 and TF Bind 10, making the optimization process more complex. Furthermore, the simplistic encoding-decoding strategy in design-bench may not accurately capture the intricacies of the neural network's accuracy, which can only be determined after training on CIFAR10.

**Summary.** ICT attains the highest rankings with a mean of $3.1/15$ and median of $2/15$ as shown in Table 2 and Figure 3, and also secures top performances in 6 out of the 7 tasks. We have further run a Welch's t-test between our method and the second-best method, obtaining p-values of $0.437$ on SuperC, $0.004$ on Ant, $0.009$ on D'Kitty, $0.014$ on Hopper, $0.000$ on TF8, $0.045$ on TF10, $0.490$ on NAS. This demonstrates statistically significant improvement in 5 out of 7 tasks, reaffirming the effectiveness of our method.

## 4.5 Ablation Studies

To better understand the impact of pseudo-label-driven co-teaching (co-teaching) and meta-learning-based sample reweighting (reweighting) on the performance of our proposed ICT method, we conduct ablation studies by removing either co-teaching or reweighting from the full ICT approach. Table 3 presents the results. Beyond

Table 3: Ablation studies on two core steps of ICT.

| Task | D | ICT | w/o co-teaching | w/o reweighting |
|------|------|------|------|------|
| TF8 | 8 | $\mathbf{0.958 \pm 0.008}$ | $0.905 \pm 0.042$ | $0.910 \pm 0.024$ |
| TF10 | 10 | $\mathbf{0.691 \pm 0.023}$ | $0.653 \pm 0.018$ | $0.654 \pm 0.023$ |
| NAS | 64 | $0.667 \pm 0.091$ | $\mathbf{0.779 \pm 0.071}$ | $0.666 \pm 0.090$ |
| SuperC | 86 | $\mathbf{0.503 \pm 0.017}$ | $0.500 \pm 0.017$ | $0.501 \pm 0.017$ |
| Ant | 60 | $\mathbf{0.961 \pm 0.007}$ | $0.927 \pm 0.033$ | $0.914 \pm 0.015$ |
| D'Kitty | 56 | $\mathbf{0.968 \pm 0.020}$ | $0.962 \pm 0.021$ | $0.959 \pm 0.013$ |
| Hopper | 5126 | $\mathbf{2.104 \pm 0.357}$ | $1.453 \pm 0.734$ | $1.509 \pm 0.166$ |

just assessing these performance indicators, we also verify the accuracy of the samples chosen by co-teaching, as well as the efficacy of the sample weights we have calculated. We do this by referring to the ground truth, with further details provided in Appendix A.4. Our reweighting module is also compared with the recently proposed RGD method [26] as detailed in the Appendix A.5.

For two of the discrete tasks (TF8 and TF10), the ICT method consistently exceeds the performance of both its ablated versions. This highlights the efficacy of the two steps when handling discrete tasks. Conversely, the exclusion of the co-teaching in NAS leads to an increase in performance. This could be attributed to the fact that the encoding-decoding strategy of NAS in design-bench is unable to accurately capture the inherent complexity of neural networks. As such, the co-teaching step, reliant on this strategy, might not be as effective. For the continuous tasks (SuperC, Ant, D'Kitty, and Hopper), we observe that the complete ICT method consistently achieves superior performance. This underlines the effectiveness of the two steps when dealing with continuous tasks. The performance gains are particularly substantial in the Hopper task when the complete ICT method is compared with the ablated versions, illustrating the power of the two steps in managing high-dimensional continuous tasks. Overall, our ablation studies demonstrate that the inclusion of both co-teaching and reweighting in the ICT method generally enhances performance across diverse tasks and input dimensions, underscoring their integral role in our approach.

## 4.6 Hyperparameter Sensitivity

We first assess the robustness of our ICT method by varying the number of samples ($K$) selected during the co-teaching process on the continuous D'Kitty Morphology task. For this analysis, $K$ is varied among $K = 8, 16, 32, 64$. In Figure 4 (a), we illustrate the $100^{th}$ percentile normalized ground-truth score as a function of time step $T$, for each of these $K$ values. The results demonstrate that the performance of ICT is resilient to variations in $K$, maintaining performances within a certain range. Additionally, ICT is capable of generating high-scoring designs early on in the process, specifically achieving such designs around the time step $t = 50$, and sustains this performance thereafter, demonstrating its robustness against the number of optimization steps $T$.

We further evaluate the robustness of our ICT method against the learning rate ($\beta$) for the meta-learning framework. As depicted in Figure 4 (b), ICT's performance remains relatively consistent across a variety of $\beta$ values, further demonstrating ICT's robustness with respect to the hyperparameter $\beta$. We explore the fine-tuning learning rate $\alpha$ and conduct further experiments and analysis on TF Bind 8. Details can be found in Appendix A.6.

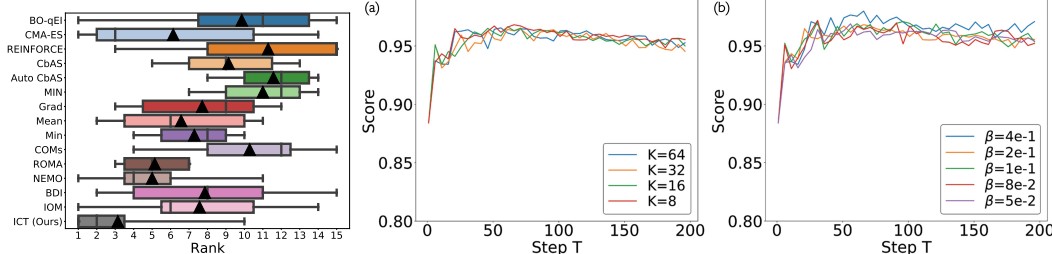

Figure 3: The whiskers in the plot mark the rank minima and maxima, whereas the vertical lines and black triangles respectively represent the median and mean.

Figure 4: Ground-truth score of the design evolves as a function of step $T$, under different values of $K$ (subplot (a)) and $\beta$ (subplot (b)), in the D'Kitty task. These results indicate that ICT attains optimal designs swiftly and maintains them stably, as well as exhibits robustness to various choices of $K$ and $\beta$.

## 5 Related Works

**Offline Model-based Optimization.** Contemporary offline model-based optimization methods can be generally classified into two primary groups: (i) generating novel designs through generative models, and (ii) conducting gradient ascent on existing designs. The former methods learn and sample from the distribution of high-scoring designs including MIN [18], CbAS [19], Auto.CbAS [20] and BootGen [27]. Recently, gradient-based methods have gained popularity due to their ability to leverage deep neural networks (DNNs) for improved design generation. These methods apply regularization techniques to either the proxy itself [8–10] or the design under consideration [11, 12], enhancing the proxy's robustness and generalization capabilities. An interesting subfield of offline MBO includes biological sequence design, which has potential applications such as designing drugs for treating diseases [27, 28]. In particular, the work [27] also adopts a proxy as a pseudo-labeler and aligns the generator with the proxy, a technique that resonates with our method. ICT falls under this category, but adopts a unique approach to improve proxy performance: it incorporates valuable knowledge from a pseudo-labeled dataset into other proxies for fine-tuning, thereby enhancing the ensemble performance. Notably, while the concurrent work of parallel mentoring [29] also employs pseudo-labeling, it focuses on pairwise comparison labels, potentially sacrificing some information due to its discrete nature.

**Sample Reweighting.** Sample reweighting is commonly utilized to address the issue of label noise [30, 31], where each sample is assigned a larger weight if it is more likely to be accurate, using a carefully designed function. Recent studies [32–34] suggest using a meta-set to guide the learning of sample weights, which can enhance model training. Such an approach is grounded in a meta-learning framework which can be used to learn hyperparameters [35–37, 34, 38–43]. Inspired by distributionally robust optimization, recent work [26] proposes a re-weighted gradient descent algorithm that provides an efficient and effective means of reweighting. In this paper, the pseudo-labeled dataset generated by co-teaching may still contain some inaccuracies, while the offline dataset is generally accurate. We propose a sample reweighting framework to reduce the inaccuracies in the pseudo-labeled dataset by leveraging the supervision signals from the offline dataset.

**Co-teaching.** Co-teaching [13] is an effective technique for mitigating label noise by leveraging insights from peer networks. It involves the concurrent training of two proxies where one proxy identifies small-loss samples within a noisy mini-batch for fine-tuning the other. Co-teaching bears similarities to decoupling [44] and co-training [45], as they all involve the interaction between two models to enhance the training process. In this study, we adapt co-teaching to work with a pseudo-labeled dataset generated by a trained proxy, instead of relying on a noisy original dataset. Specifically, we employ one proxy to select accurate samples from this pseudo-labeled dataset for fine-tuning the other, and vice versa.

## 6 Conclusion and Discussion

In this study, we introduce the ICT (Importance-aware Co-Teaching) method for mitigating the out-of-distribution issue prevalent in offline model-based optimization. ICT is a two-step approach. The first step is pseudo-label-driven co-teaching, which iteratively selects a proxy to generate pseudo-labeled data. Valuable data are identified by co-teaching to fine-tune other proxies. This process,

repeated three times with different pseudo-labelers, facilitates knowledge transfer. In the second step, meta-learning-based sample reweighting assigns and updates importance weights to samples selected by the co-teaching process, further improving the proxy fine-tuning. Our experimental findings demonstrate the success of ICT. We discuss its limitations in Appendix A.7

**Future Work.** Though we initially design ICT with three proxies, the method's inherent scalability and flexibility make it applicable to scenarios involving $N$ proxies. In such a scenario, we can iteratively select one proxy out of $N$ as the pseudo-labeler to generate data. Then, each of the remaining $N - 1$ proxies could select small-loss samples from its perspective and provide these samples to the other $N - 2$ proxies for fine-tuning. This process enhances knowledge transfer and facilitates cooperative learning among the proxies. Looking to the future, we plan to conduct further research into the dynamics of such an expanded ensemble of proxies.

**Negative Impact.** It is crucial to recognize that ICT's potential benefits come with possible negative consequences. Advanced optimization techniques can be applied for both constructive and destructive purposes, depending on their use. For example, while drug development and material design can have a positive impact on society, these techniques could also be misused to create harmful substances or products. As researchers, we must remain attentive and strive to ensure that our work is employed for the betterment of society while addressing any potential risks and ethical concerns.

## 7   Acknowledgement

This research was empowered in part by the computational support provided by Compute Canada (www.computecanada.ca).

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

# A  Appendix

## A.1  Considerations for Sampling Around the Offline Dataset

In this subsection, we explore an alternative sampling strategy for the pseudo-labeling process. Instead of generating new samples around the current optimization point, this strategy generates samples directly around the offline dataset $\mathcal{D}$. To ascertain the effectiveness of our chosen strategy against this alternative, we perform experiments on two tasks: D'Kitty (continuous) and TF8 (discrete).

Table 4 showcases the results. For both tasks, our strategy consistently yields higher scores, affirming its superior performance over the alternative. The advantage of our chosen strategy can be attributed to its dynamic nature. By sampling around the current optimization point, we gather more insightful information for the local fine-tuning of the proxy. This strategy allows the co-teaching process to adapt and evolve according to the optimization trajectory, leading to improved performances.

Table 4: Comparison of Sampling Strategies.

| Method | Sampling along Gradient Path (**Ours**) | Sampling from $\mathcal{D}$ |
|--------|------------------------------------------|------------------------------|
| TF8 | **0.958 ± 0.008** | 0.871 ± 0.067 |
| D'Kitty | **0.968 ± 0.020** | 0.955 ± 0.006 |

## A.2  Comparative Performance Analysis using Median Scores

In addition to the maximum scores discussed in the main paper, we also present the median ($50^{th}$ percentile) scores across all seven tasks. The best design in the offline dataset, denoted as $\mathcal{D}(\textbf{best})$, along with the mean and median rankings are provided for comprehensive comparison.

**Performance in Continuous Tasks.** Table 5 illustrates the performances of ICT compared with other methods in continuous tasks. It is noteworthy that ICT exhibits performance on par with the best-performing methods. Compared with the vanilla gradient ascent (Grad), ICT demonstrates superior performance, thus affirming its effectiveness in addressing out-of-distribution issues. Moreover, ICT is generally better than the mean ensemble (Mean), which demonstrates the effectiveness of our strategy. These results support the use of ICT as a robust baseline for offline MBO.

Table 5: Experimental results on continuous tasks for comparison (median).

| Method | Superconductor | Ant Morphology | D'Kitty Morphology | Hopper Controller |
|--------|----------------|----------------|---------------------|--------------------|
| $\mathcal{D}(\textbf{best})$ | 0.399 | 0.565 | 0.884 | 1.0 |
| BO-qEI | 0.300 ± 0.015 | 0.567 ± 0.000 | **0.883 ± 0.000** | 0.343 ± 0.010 |
| CMA-ES | 0.379 ± 0.003 | −0.045 ± 0.004 | 0.684 ± 0.016 | −0.033 ± 0.005 |
| REINFORCE | **0.463 ± 0.016** | 0.138 ± 0.032 | 0.356 ± 0.131 | −0.064 ± 0.003 |
| CbAS | 0.111 ± 0.017 | 0.384 ± 0.016 | 0.753 ± 0.008 | 0.015 ± 0.002 |
| Auto.CbAS | 0.131 ± 0.010 | 0.364 ± 0.014 | 0.736 ± 0.025 | 0.019 ± 0.008 |
| MIN | 0.336 ± 0.016 | **0.618 ± 0.040** | **0.887 ± 0.004** | 0.352 ± 0.058 |
| Grad | 0.321 ± 0.010 | 0.559 ± 0.032 | 0.856 ± 0.009 | 0.354 ± 0.010 |
| Mean | 0.334 ± 0.003 | 0.569 ± 0.010 | 0.876 ± 0.003 | 0.386 ± 0.003 |
| Min | 0.354 ± 0.026 | 0.571 ± 0.011 | **0.883 ± 0.000** | 0.359 ± 0.004 |
| COMs | 0.316 ± 0.026 | 0.560 ± 0.002 | 0.879 ± 0.002 | 0.341 ± 0.009 |
| ROMA | 0.372 ± 0.019 | 0.479 ± 0.041 | 0.853 ± 0.007 | 0.389 ± 0.005 |
| NEMO | 0.318 ± 0.008 | **0.592 ± 0.000** | 0.880 ± 0.000 | 0.355 ± 0.002 |
| BDI | 0.412 ± 0.000 | 0.474 ± 0.000 | 0.855 ± 0.000 | **0.408 ± 0.000** |
| IOM | 0.352 ± 0.021 | 0.509 ± 0.033 | 0.876 ± 0.006 | 0.370 ± 0.009 |
| **ICT**$_{\text{(ours)}}$ | 0.399 ± 0.012 | **0.592 ± 0.025** | 0.874 ± 0.005 | 0.362 ± 0.004 |

**Performance in Discrete Tasks.** The median scores for discrete tasks are reported in Table 6. ICT consistently demonstrates high performance for both TF Bind 8 and TF Bind 10. However, for the NAS task, which has a higher dimensionality than the two tasks, the optimization process becomes notably more complex. Further, the simplistic encoding-decoding strategy employed in the design bench may not accurately capture the intricacies of the neural network's accuracy, potentially contributing to ICT's suboptimal performance on the NAS task.

Table 6: Experimental results on discrete tasks & ranking on all tasks for comparison (median).

| Method | TF Bind 8 | TF Bind 10 | NAS | Rank Mean | Rank Median |
|---|---|---|---|---|---|
| $\mathcal{D}(\textbf{best})$ | 0.439 | 0.467 | 0.436 | | |
| BO-qEI | $0.439 \pm 0.000$ | $0.467 \pm 0.000$ | $0.544 \pm 0.099$ | 7.7/15 | 8/15 |
| CMA-ES | $0.537 \pm 0.014$ | $0.484 \pm 0.014$ | $\textbf{0.591} \pm \textbf{0.102}$ | 8.4/15 | 6/15 |
| REINFORCE | $0.462 \pm 0.021$ | $0.475 \pm 0.008$ | $-1.895 \pm 0.000$ | 10.9/15 | 14/15 |
| CbAS | $0.428 \pm 0.010$ | $0.463 \pm 0.007$ | $0.292 \pm 0.027$ | 12.9/15 | 13/15 |
| Auto.CbAS | $0.419 \pm 0.007$ | $0.461 \pm 0.007$ | $0.217 \pm 0.005$ | 13.4/15 | 13/15 |
| MIN | $0.421 \pm 0.015$ | $0.468 \pm 0.006$ | $0.433 \pm 0.000$ | 7.7/15 | 9/15 |
| Grad | $0.528 \pm 0.021$ | $0.519 \pm 0.017$ | $0.438 \pm 0.110$ | 7.7/15 | 8/15 |
| Mean | $0.539 \pm 0.030$ | $\textbf{0.539} \pm \textbf{0.010}$ | $0.494 \pm 0.077$ | 5.3/15 | 5/15 |
| Min | $\textbf{0.569} \pm \textbf{0.050}$ | $0.485 \pm 0.021$ | $\textbf{0.567} \pm \textbf{0.006}$ | $\textbf{3.7/15}$ | 4/15 |
| COMs | $0.439 \pm 0.000$ | $0.467 \pm 0.002$ | $0.525 \pm 0.003$ | 8.4/15 | 8/15 |
| ROMA | $\textbf{0.555} \pm \textbf{0.020}$ | $0.512 \pm 0.020$ | $0.525 \pm 0.003$ | 5.6/15 | 5/15 |
| NEMO | $0.438 \pm 0.001$ | $0.454 \pm 0.001$ | $\textbf{0.564} \pm \textbf{0.016}$ | 7.7/15 | 7/15 |
| BDI | $0.439 \pm 0.000$ | $0.476 \pm 0.000$ | $0.517 \pm 0.000$ | 6.7/15 | 8/15 |
| IOM | $0.439 \pm 0.000$ | $0.477 \pm 0.010$ | $-0.050 \pm 0.011$ | 7.9/15 | 7/15 |
| $\textbf{ICT}_{\text{(ours)}}$ | $\textbf{0.551} \pm \textbf{0.013}$ | $\textbf{0.541} \pm \textbf{0.004}$ | $0.494 \pm 0.091$ | 4.3/15 | $\textbf{3/15}$ |

**Summary.** ICT excels by achieving the best median ranking and a top-two mean ranking. These rankings consolidate ICT's standing as a strong method for both continuous and discrete tasks.

## A.3 Hyperparameter Setting

We report the details of hyperparameters in our experiments. The number of iterations, $T$, is set to 200 for continuous tasks and 100 for discrete tasks. For most continuous and discrete tasks, we employ the Adam optimizer [46] to fine-tune the proxies. The learning rates are set at $1e-3$ and $1e-1$ for continuous tasks and discrete tasks, respectively. In the case of the Hopper Controller task, the input dimension is significantly larger, at $5126$, and we adopt a smaller learning rate $1e-4$ for fine-tuning to ensure stability of the optimization process. Regarding the learning rate for the meta-learning framework, we use the Adam optimizer [46] with a learning rate $2e-1$ for continuous tasks and $3e-1$ for discrete tasks, respectively.

## A.4 Analysis of Co-teaching and Sample Reweighting Efficacy

In our analysis, we focus on two key steps of our method: (1) pseudo-label-driven co-teaching and (2) meta-learning-based sample reweighting. We evaluate the efficacy of these steps by comparing generated samples with their corresponding ground truth. It's important to note that during the training phase, ground-truth scores are inaccessible to all algorithms and are used strictly for evaluation. Our method incorporates three proxies $f_{\boldsymbol{\theta}_1}(\cdot)$, $f_{\boldsymbol{\theta}_2}(\cdot)$, and $f_{\boldsymbol{\theta}_3}(\cdot)$. We employ $f_{\boldsymbol{\theta}_1}(\cdot)$ for pseudo-labeling and $f_{\boldsymbol{\theta}_2}(\cdot)$, $f_{\boldsymbol{\theta}_3}(\cdot)$ for co-teaching. We run ICT over 50 time steps for both D'Kitty (continuous) and TF8 (discrete) tasks.

**Pseudo-label-driven co-teaching.** The step involves selecting 64 samples with smaller losses for fine-tuning the proxies while ignoring the remaining 64 samples. To assess the effectiveness of this strategy, we calculate $\mathcal{L}^{Sel}$, the mean squared error (MSE) between the pseudo-labeled and ground truth scores of the selected 64 samples, and $\mathcal{L}^{Ign}$, the MSE for the ignored samples. These calculations are averaged over 50 steps. We find that for D'Kitty, $\mathcal{L}^{Sel}$ is 0.124 lower than $\mathcal{L}^{Ign}$ and for TF8, it's 0.006 less than $\mathcal{L}^{Ign}$. These results validate the efficacy of this step, as the selected samples more closely align with the ground truth.

**Meta-learning-based sample reweighting.** In this step, we aim to assign larger weights to cleaner samples and smaller weights to noisier ones among the total of 64 samples. We measure the efficacy of this step by calculating $\mathcal{L}^{Large}$, the MSE between the pseudo-labeled and ground-truth scores of the 32 samples with larger weights, and $\mathcal{L}^{Small}$, the MSE for the 32 samples with smaller weights. These calculations are averaged over 50 steps. We observe that for D'Kitty, $\mathcal{L}^{Large}$ is 0.010 lower than $\mathcal{L}^{Ign}$. For TF8, $\mathcal{L}^{Large}$ is 0.005 less than $\mathcal{L}^{Small}$. These findings indicate that the samples with larger weights are indeed closer to the ground truth, substantiating the effectiveness of this step.

## A.5   Comparison with RGD

We conduct a deeper analysis comparing our sample reweighting method with the RGD method, as proposed in [26]. The RGD method, drawing inspiration from distributionally robust optimization, aims to provide an efficient and theoretically sound way to reweight samples during training.

To ensure a fair comparison, we perform additional experiments, keeping the dataset and experimental conditions consistent for both methods. Two benchmarks, TFB8 and D'Kitty, are evaluated.

Table 7: Comparison between Sample Reweighting (Ours) and RGD-EXP.

| Method | TFB8 | D'Kitty |
|---|---|---|
| Sample Reweighting (Ours) | $0.958 \pm 0.008$ | $0.968 \pm 0.020$ |
| RGD-EXP | $0.906 \pm 0.058$ | $0.955 \pm 0.014$ |

Our results in Table 7 indicate that while RGD is efficient and provides a theoretically effective solution, our approach delivers slightly superior performance. This performance edge might stem from our method's capability to utilize information from the static dataset.

## A.6   Examining Hyperparameter Sensitivity Further

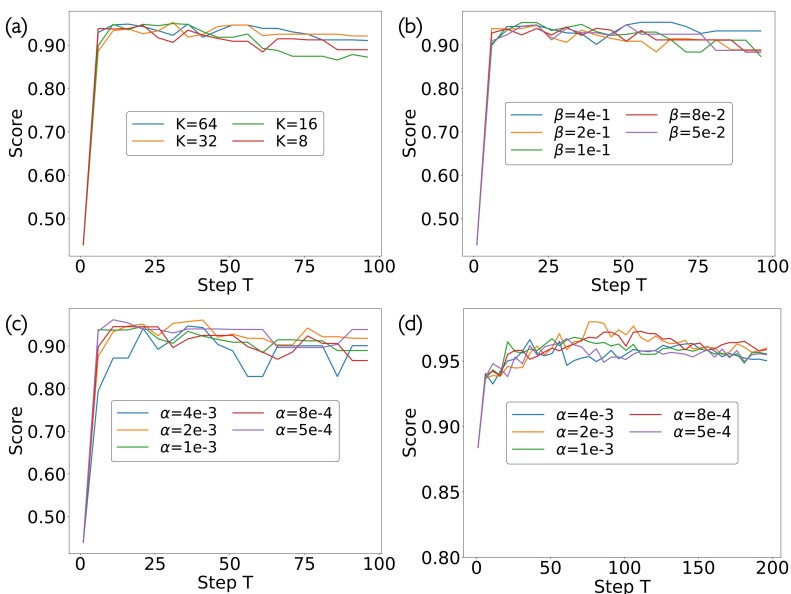

Figure 5: Extended Analysis on Hyperparameter Sensitivity.

Building on the analysis from Sec 4.6, we delve deeper into hyperparameter sensitivity, focusing on the TF8 task. Specifically, we investigate the influence of the number of selected samples ($K$) in the first step, and the learning rate ($\beta$) in the second step.

- Figure 5 (a) displays the $100^{th}$ percentile normalized ground-truth score as a function of the time step $T$ for different $K$ values $(8, 16, 32, 64)$. ICT demonstrates stability over a specific range for varying $K$ values, showcasing its robustness. Notably, ICT reaches optimal designs around $t = 20$ and maintains this level, further validating its resilience against different optimization steps $T$.

- Figure 5 (b) plots the $100^{th}$ percentile normalized ground-truth score as a function of the learning rate ($\beta$) in TF8. ICT maintains a consistent performance across diverse $\beta$ values, corroborating its robustness concerning the hyperparameter $\beta$ in TF8.

Furthermore, we evaluate the effect of the fine-tuning learning rate $\alpha$ in both TF8 and D'Kitty tasks. Figures 5 (c) and 5 (d) reveal a consistent performance across varied $\alpha$ values for both tasks, highlighting ICT's robustness towards the fine-tuning learning rate.

## A.7 Limitation

We validate the effectiveness of ICT across a broad spectrum of tasks. Nevertheless, certain evaluation methodologies do not completely represent authentic situations. For instance, in the superconductor task [5], we adhere to the established convention of utilizing a random forest regression model as the oracle, in line with previous studies [1]. Regrettably, this model may not perfectly mirror the complexities of real-world cases, resulting in discrepancies between our oracle and the ground-truth. Future collaborations with domain experts can potentially refine these evaluation methods. Overall, given the straightforward formulation of ICT, combined with empirical proof of its robustness and effectiveness across diverse tasks in the design-bench [1], we maintain confidence in its capability to effectively generalize to other scenarios.

