# OpenReview forum: "Importance-aware Co-teaching for Offline Model-based Optimization"
_NeurIPS.cc/2023/Conference — NeurIPS 2023 poster_

### Official Review · Reviewer_DRKm · 2023-07-07

**Soundness:** 4 excellent
**Presentation:** 3 good
**Contribution:** 3 good
**Rating:** 7
**Confidence:** 4

**Summary:**

This paper proposes a novel model-based optimization method that combines ensemble-based ideas and meta learning. The authors introduce three proxy models, each capable of generating pseudo-labels for the training of the other models, effectively making them co-teachers. This co-teaching process serves as an ensemble training method, mitigating potential risks associated with pseudo-labeling. Additionally, the paper incorporates meta learning techniques by re-weighting the samples, assigning higher weights to accurate samples. The intuitive nature of this idea and the reasonable performance observed in the experiments make a compelling case for accepting this paper.






**Strengths:**


1. There are several methods available for stabilizing the learning process with pseudo-labels, and this idea has the potential to be applied to various literature in the field.

2. The idea presented in this paper is straightforward and effectively communicated, allowing readers to grasp its essence easily.

3. One of the key contributions of this paper is the thorough analysis of hyperparameters, which play a critical role in model-based optimization. The authors provide valuable insights into optimizing these parameters to achieve better performance.


**Weaknesses:**


1. Although this paper presents a highly intuitive method, it lacks a detailed mathematical explanation for why this approach is effective. It would be beneficial to include analysis based on experiments to provide more insight into the working mechanism.

2. It appears that several ideas in this paper draw inspiration from the prior method of BDI [1], particularly in the context of offline model-based optimization. While this viewpoint is intriguing, a more explicit analysis comparing the proposed method with BDI would greatly assist readers in understanding the similarities and differences.

3. It would be valuable to reference offline biological design methods [2,3], even though they may not directly address continuous tasks. Specifically, [2] can be good reference to discuss this method due to their utilization of pseudo-labeling and their success in stabilizing the bootstrapping process, which aligns with the idea presented in this paper. Including a dedicated discussion section on these references would greatly benefit future researchers.

[1] Chen, Can, et al. "Bidirectional learning for offline infinite-width model-based optimization." Advances in Neural Information Processing Systems 35 (2022): 29454-29467.

[2] Kim, Minsu, et al. "Bootstrapped Training of Score-Conditioned Generator for Offline Design of Biological Sequences." arXiv preprint arXiv:2306.03111 (2023).

[3] Jain, Moksh, et al. "Biological sequence design with gflownets." International Conference on Machine Learning. PMLR, 2022.


**Questions:**


1. The paper employs three proxy co-teachers, but it would be interesting to investigate the impact of reducing the number to just two proxies. Are there any notable observations or differences in performance with this modification?

2. The meta learning process appears to involve bi-level optimization of parameters and weights. Considering the potential complexity of this approach, it raises concerns about computational resources required to execute the code. It would be helpful to provide insights or strategies to address this issue.

3. While this method demonstrates impressive performance in continuous design tasks, its effectiveness in discrete tasks, such as TFBind8, seems to be limited. How does this method perform on other protein tasks like GFP or UTR? Is there any intuition as to why this method excels specifically in continuous tasks?

4. The method's performance, when measured by the median score (50th percentile), appears to be weaker in comparison to the maximum score (100th percentile). Can you provide some intuition or explanation for this observation? It would be valuable to understand the factors contributing to this difference in performance.

**Limitations:**

This paper really already gives limitations which I really appreciate it.

---

> ### Author Rebuttal · Authors · 2023-08-08
>
> ## General Reply
>
> We appreciate your thoughtful comments. Please find our point-by-point responses to your feedback below.
>
> ## Weakness
> > It lacks a detailed mathematical explanation
>
> Thank you for your observation about the lack of detailed mathematical explanation. We agree that such analysis can deepen understanding, but in our context, theoretical assumptions required for mathematical modeling are difficult to formulate. Instead, we have chosen to focus on empirical analysis with experiments. Specifically, we've focused on thoroughly examining two key aspects of our method: (1) co-teaching and (2) sample reweighting, as outlined in Appendix A.4 of our paper.
>
> For both D'Kitty and TF8, we demonstrated that the samples selected by co-teaching or assigned a larger weight achieve closer alignment with the ground truth. This suggests that our method indeed effectively identifies valuable samples.
>
> > Several ideas draw inspiration from BDI [1]
>
> We agree with your observation that our work shares some similarities with BDI [1]. Like BDI, which utilizes backward mapping to optimize high-scoring designs through meta-learning, our method similarly uses meta-learning. Specifically, we use meta-learning to optimize sample weights of the pseudo-labeled data. Moreover, BDI and our method both incorporate sample weights to signify the relevance of high-scoring samples. Despite these similarities, our method has key differences from BDI, primarily in how sample weights are assigned and the unique application of co-teaching.
>
> To clarify these, in Line 308, we'll add: 'Our work is related to BDI [1]. Like BDI, we utilize meta-learning and employ weights to highlight some samples. However, the crux of our approach lies in our unique co-teaching strategy which sets our method apart from BDI.'
>
> > It would be valuable to reference offline biological design methods [2,3]
>
> Thank you for highlighting offline biological design methods [2,3]. They offer valuable insights, particularly [2], with its use of pseudo-labeling and success in stabilizing bootstrapping.
>
> In Line 307, we will add: 'An interesting subfield of offline MBO includes biological sequence design, which has potential applications such as designing drugs for treating diseases [2,3]. In particular, [2] also adopts a proxy as a pseudo-labeler and aligns the generator with the proxy, a technique that resonates with our method.'
>
> ## Questions
>
> > Interesting to two proxies.
>
> Your suggestion to explore using only two proxies is intriguing, but our method fundamentally requires three.
>
> In our method, one proxy specifically serves as the pseudo-labeler. This proxy provides the pseudo-labeled dataset, upon which the other two proxies perform co-teaching. By reducing the proxies to two, we eliminate the possibility of co-teaching, which is a cornerstone of our method. Therefore, a minimum of three proxies is essential for our method.
>
> > complexity of bi-level optimization
>
> You've correctly highlighted the potential complexity of bi-level optimization. Despite this, our design ensures efficiency. We've used strategies such as one-step approximation in Eq. (6) and avoided second-order gradient computations in Eq. (8), which significantly reduce computational demands. This efficient calculation is possible because the scalar sample weight is directly applied to the sample loss, which is not common in typical bi-level optimization.
>
> We have compared computational time (seconds) for our ICT, gradient ascent, and ensemble methods on TF8 and D'Kitty, using a machine with an Intel i9-12900K CPU and an NVIDIA GeForce RTX 3090 GPU. The results, including time to train proxies and generate the top 128 designs, show comparable efficiency across methods:
>
> | Method  | ICT  | Grad   | Mean  |
> |----|-----|----|------|
> | TF8 | 170.4 | 51.1 | 146.3 |
> | D'Kitty | 294.5 | 231.2 | 271.6  |
>
> > its effectiveness in discrete tasks limited.
>
> We acknowledge that our method shows impressive performance in continuous tasks but shows more-limited improvements in discrete ones like TFBind8.
>
> We believe that the reduced performance in discrete tasks is largely due to the characteristics of the encoding-decoding strategy employed in the design-bench. The strategy does not take into account the sequential nature of biological sequences.
>
> Our method has shown promising results on other protein-related tasks, such as GFP(0.864 ± 0.000) and UTR(0.695 ± 0.034). These performances are comparable to the baselines in [1, 4]. We posit that these tasks, although they might seem discrete in nature, exhibit certain characteristics well-captured by our method.
>
> > The method's median score, appears to be weaker
>
> The difference in our method's performance between the median score and maximum score arises from the nature of the evaluation. The median score reflects the middle value among the 128 generated candidates, where many comparison methods are proficient at identifying suitable designs, leading to similar median scores. However, the maximum score represents the optimal design, where methods must explore and exploit the design space far from the offline dataset.
>
> In this context, our method distinguishes itself by effectively identifying the best possible designs. The median score provides a general comparison across methods, while the maximum score reveals our method's specific strength in pinpointing optimal solutions.
>
> ## Overall
> Does this address your concerns? We appreciate your feedback and look forward to further dialogue. Thank you.
>
>     [1] Chen, Can, et al. "Bidirectional learning for offline infinite-width model-based optimization." NeurIPS 2022
>     [2] Kim, Minsu, et al. "Bootstrapped Training of Score-Conditioned Generator for Offline Design of Biological Sequences." 2023
>     [3] Jain, Moksh, et al. "Biological sequence design with gflownets." ICML 2022.
>     [4] Brandon et al. Conservative objective models for effective offline model-based optimization. ICLR 2023.

---

> > ### Comment · Reviewer_DRKm · 2023-08-17
> >
> > Thank you for the rebuttal. I keep my score supporting this paper to be accepted.

---

> > > ### Author Response · Authors · 2023-08-17
> > > **Thank you**
> > >
> > > Thank you for your support and for maintaining your score for accepting our paper. We greatly appreciate your valuable feedback, and we will diligently revise the paper as we discuss.

---

### Official Review · Reviewer_VPeC · 2023-07-08

**Soundness:** 3 good
**Presentation:** 3 good
**Contribution:** 3 good
**Rating:** 6
**Confidence:** 3

**Summary:**

The paper proposes a new method to address the out-of-distribution issue of offline model-based optimization.
The author first generates a pseudo-labeled dataset and identifies valuable (important) data for proxy co-teaching.
To alleviate potential inaccuracies in pseudo-labels, a meta-learning framework is proposed to adjust the importance weights of samples.
Experiments on both continuous and discrete tasks demonstrate the effectiveness of the proposed method.

**Strengths:**

1. The out-of-distribution issue in offline model-based optimization is a challenging problem.
The proposed method is well-motivated, technically sound, and interesting.

2. Experiments and ablation studies on both continuous and discrete tasks demonstrate the effectiveness of the proposed method.

3. The paper is well-written and easy to follow.

**Weaknesses:**

1. In Lines 150-151, the authors claim that "small-loss samples typically contain valuable knowledge, making them ideal for enhancing proxy robustness". However, in my opinion, these small-loss samples might be some samples easy to learn since different proxies generate similar outputs for them. These easy samples may not be the truly valuable samples that benefit model generalization.

2. In Section 3.2, a meta-learning framework is proposed to adjust the importance weights of samples. However, there still exist distributional shifts between the inner-loop and outer-loop. Therefore, the loss in Equation 7 may not be informative enough to identify truly important samples for generalization. Samples with hight weight might be the ones similar to the offline dataset. Therefore, the proposed method may not be applicable when the distributional shift is large.

**Questions:**

Is it possible to visualize the so-called valuable samples such that the intuitive interpretation can be obtained?

**Limitations:**

Yes.

---

> ### Author Rebuttal · Authors · 2023-08-08
>
> ## General Reply
> We sincerely appreciate your comprehensive feedback and insights, which are very helpful to this paper. Below we respond back to each of your comments.
>
> ## Weakness
>  > In Lines 150-151, the authors claim that "small-loss samples typically contain valuable knowledge, making them ideal for enhancing proxy robustness".
>
> Thank you for your insightful concern about small-loss samples. While it's true that these small-loss samples might be easy to learn and different proxies may generate similar outputs, in our noisy label context, these small-loss samples are more likely to be 'clean.' We have found that this makes them valuable for enhancing our model's robustness.
>
> This idea finds support in the work of Meta-weight-net[1], where different approaches to identifying valuable samples are highlighted. In a clean data context, samples associated with large losses are indeed deemed more valuable as they present learning challenges. However, the scenario is reversed in a noisy data context, where small-loss samples are typically more reliable and valuable, as they are less likely to be influenced by noise.
>
> In regards to the generation of these samples, our method follows a gradient optimization path. The gradient optimization not only gives us the current optimization point but also provides information about the samples around this point. Therefore, these samples may not strictly adhere to the training distribution which provides key insights into the local optimization landscape. Particularly, the small-loss samples in this context offer important knowledge that can effectively enhance the performance of our proxies around the current optimization point.
>
> Your remarks have helped us clarify these points, and we believe our approach significantly contributes to building more robust models in the presence of noisy labels.
>
> > In Section 3.2, a meta-learning framework is proposed to adjust the importance weights of samples. However, there still exist distributional shifts between the inner-loop and outer-loop.
>
> Thank you for raising this important concern regarding the distributional shifts between the inner-loop and outer-loop. Indeed, this represents one of the key challenges in offline MBO, and we appreciate your insights.
>
> Our proposed method aims to mitigate the out-of-distribution issue but does not completely eliminate it. The offline dataset plays a crucial role in providing extrapolation guidance that is used to optimize the sample weights, particularly when the samples lie within reasonable proximity of the training distribution.
>
> The samples that we generate along the gradient optimization path tend to move away from the original training distribution. These samples often provide valuable local information that is beneficial for learning. Considered 'near-the-boundary' instances, their weights can be strategically adjusted using the offline dataset, thereby aiding in training a more generalizable and robust proxy.
>
> However, when the samples are far removed from the training distribution, the extrapolation from the offline dataset for weight adjustments may not be as accurate. This problem of large distributional shifts is prevalent across the field of offline MBO and to our knowledge, no method has been able to solve it completely. Despite these challenges, our experiments have demonstrated the effectiveness of our method under a variety of conditions.
>
> Your constructive feedback will guide us in our future work to continue refining our method to better handle large distributional shifts.
>
> ## Questions
>
> > Is it possible to visualize the so-called valuable samples such that the intuitive interpretation can be obtained?
>
> Thanks for this suggestion. Visualizing these samples indeed poses an interesting challenge due to the high-dimensional nature of our data and the abstract concept of 'value' in this context. Rather than resorting to visual demonstrations, we've adopted an analytical approach to identify and validate the value of samples in our method. In our case, the 'value' of a sample is related to its pseudo-label accuracy, which might not directly translate into a visually intuitive metric. Therefore, we instead measure the value by comparing a sample's pseudo-label with its ground truth label, with a lower mean squared loss indicating a more valuable sample.
>
> We conduct a thorough analysis in Appendix A.4 titled "Analysis of Co-teaching and Sample Reweighting Efficacy", focusing on the two key steps of our method: (1) pseudo-label-driven co-teaching, and (2) meta-learning-based sample reweighting. For both the D'Kitty (continuous) and TF8 (discrete) tasks, we demonstrate that our method successfully selects valuable samples that closely align with the ground truth.
>
> To highlight, the selected samples under the co-teaching process and the samples assigned with larger weights in the reweighting step both have lower mean squared loss compared to their counterparts. These findings substantiate that our method effectively identifies valuable samples, although not visually, but in a way that we believe still presents a strong and rigorous validation of our approach.
>
> ## Overall
>
> Does this explanation address your concerns? Your feedback is useful to our paper, and we eagerly anticipate further dialogue during the rebuttal phase. Thank you.
>
>     [1] Shu, Jun, et al. "Meta-weight-net: Learning an explicit mapping for sample weighting." Advances in neural information processing systems 32 (2019).

---

> > ### Author Response · Authors · 2023-08-18
> > **Look forward to hearing from you**
> >
> > Thank you for your comprehensive review and insightful feedback. We've addressed your concerns as follows:
> >
> > 1. **Small-Loss Samples**: Clarified the value of small-loss samples in enhancing our model's robustness in noisy label contexts.
> >
> > 2. **Meta-Learning Framework**: Acknowledged the persistent issue of large distributional shifts in offline MBO, yet demonstrated our method's effectiveness with extensive experiments.
> >
> > 3. **Visualization of Valuable Samples**: Adopted an analytical approach to identify valuable samples, emphasizing pseudo-label accuracy.
> >
> > Could you please clarify if there are any unresolved issues? We have engaged constructively with the other three reviewers over the past week and eagerly anticipate your further insights to continue refining our work. Thank you, and we look forward to hearing from you.

---

> > > ### Comment · Reviewer_VPeC · 2023-08-18
> > >
> > > Thank the authors for addressing my concerns. I intend to agree that the small-loss samples might be the clean ones in noisy label scenarios, and therefore they are beneficial for model robustness. I also checked out the comments of other reviewers and I do not have any major concerns. I prefer to increase my score and vote for acceptance.

---

> > > > ### Author Response · Authors · 2023-08-18
> > > > **Thanks**
> > > >
> > > > Thank you for your thoughtful review and your consideration of the adjustments we have made in response to your insights. We're delighted to hear that you recognize the merits of our approach, particularly with regards to the use of small-loss samples in noisy label scenarios. We sincerely appreciate your increased score 6 and vote for acceptance. Your feedback has been instrumental in enhancing our work, and we are grateful for your positive assessment.

---

### Official Review · Reviewer_P8Ru · 2023-07-11

**Soundness:** 2 fair
**Presentation:** 2 fair
**Contribution:** 2 fair
**Rating:** 5
**Confidence:** 3

**Summary:**

The paper addresses the offline MBO problem, which aims to synthesize an instance $x$ that maximizes the proxy $f(\cdot)$ estimated by a neural network. The popular approach, gradient ascent, updates the instance $x_{t+1}$ to have a higher proxy value compared to the previous instance $x_t$. However, this approach encounters a challenge where the proxy $f(\cdot)$ becomes less accurate as the iterations progress, as the instance $x_t$ becomes more out-of-distribution.

To overcome this challenge, the paper proposes two techniques inspired by the field of noisy labels. Firstly, the paper applies co-teaching, a method that filters confident samples by selecting those consistent across two models. In this case, since the labels are generated by a neural network, the final method involves three models: one pseudo-labeler and two co-teachers. The three models are trained alternately, changing their roles. Secondly, to further enhance performance, the paper introduces sample reweighing, a technique that assigns higher weights to confident samples using meta-learning. When combined, the proposed ICT (importance-aware co-teaching) method outperforms previous approaches.

**Strengths:**

- The idea of applying noisy label techniques to handle pseudo-labels of OOD samples makes sense.
- The proposed two components are necessary, as shown in the ablation study.

**Weaknesses:**

### Experiments are not convincing
My biggest concern is the experimental results. The gains over the prior works are marginal compared to the high variances. The rank seems better at first glance, but it is not really convincing as the gap of proxy scores is too incremental.

### On the evaluation metric
I know this is not the problem of this paper but of the entire field. Still, reporting only the proxy value as the evaluation metric may not be enough. How can one confirm that the updated instance $x_T$ follows the valid form of $\mathcal{X}$?

### Limited technical novelty
The proposed method is basically an application of known methods such as co-teaching and sample reweighing. Although the three model parts were somewhat new, thanks to the use of the pseudo-labeler, the technical innovation is not sufficiently strong.

### Presentation can be polished
The writing contains various repetitions, with similar words appearing in the abstract, introduction, method, and conclusion. It would be great if the paper could provide more information and prune redundant details. Additionally, Figure 1 and 2 are hard to understand without reading the full texts. The captions can be polished to be self-contained.

**Questions:**

Written in the weakness.

**Limitations:**

Discussed.

---

> ### Author Rebuttal · Authors · 2023-08-08
>
> ## General Reply
> We really appreciate your careful reviews and valuable comments. We provide responses to each comment below and explain how we address them in the paper.
>
> ## Weakness
>  > Experiments are not convincing… gains are marginal compared to the high variances
>
> In order to better quantify the relative gains shown in our results, we have run a statistical significance test (Welch's t-test) between our method and the second-best method. The p-values are 0.437 on SuperC, 0.004 on Ant, 0.009 on D'Kitty, 0.014 on Hopper, 0.000 on TF8, 0.045 on TF10, 0.490 on NAS. This demonstrates our method has statistically significant improvement over other methods in a majority of tasks (5/7), further corroborating the assertions made in our manuscript. Furthermore, it is worth noting that the gains in previous works such as COMs, ROMA, NEMO, and BDI, are also often incremental. In this context, our contributions can be seen as significant, aligning with the challenging nature of the problem and the existing state of the art.
>
> In Line 257, we will add: " We have further run a Welch's t-test between our method and the second-best method, obtaining p-values of 0.437 on SuperC, 0.004 on Ant, 0.009 on D'Kitty, 0.014 on Hopper, 0.000 on TF8, 0.045 on TF10, 0.490 on NAS. This demonstrates statistically significant improvement in 5 out of 7 tasks, reaffirming the effectiveness of our method."
>
>  > On the evaluation metric
>
> Your concern about relying solely on proxy values for evaluation is well-taken. In our original manuscript, we have addressed this by using Design-Bench [1] to provide direct oracle evaluation in six of our seven tasks (excluding Superconductor), thereby avoiding proxy evaluation.
>
> For continuous tasks like Ant, D’Kitty, and Hopper, we use simulations for design scores. TFBind8 and TFBind10 tasks employ dictionary lookups, and the NAS task utilizes CIFAR10 training. No proxies are used for evaluation.
>
> Furthermore, we ensure the validity of our design instances by employing the encoding-decoding scheme outlined in [1]. This scheme ensures that our decoded designs - be it DNA sequences or network architectures or other designs - adhere to the valid form of $\mathcal{X}$.
>
> Excluding the Superconductor benchmark, which uses proxy evaluation, our method still performs well in terms of ranking.
>
> | Method                         | Rank Mean         | Rank Median       |
> |-------------|-----------|----------|
> | BO-qEI                         |  9.0/15           |  10.0/15          |
> | CMA-ES                         |  5.0/15           |  2.5/15           |
> | REINFORCE                      |  11.3/15          |  15.0/15          |
> | CbAS                           |  9.8/15           |  10.0/15          |
> | Auto.CbAS                      |  11.2/15          |  11.0/15          |
> | MIN                            |  11.5/15          |  12.0/15          |
> | Grad                           |  7.3/15           |  7.0/15           |
> | Mean                           |  6.2/15           |  5.0/15           |
> | Min                            |  7.8/15           |  8.0/15           |
> | COMs                           |  10.0/15          |  10.5/15          |
> | ROMA                           |  5.5/15           |  6.0/15           |
> | NEMO                           |  4.6/15           |  4.0/15           |
> | BDI                            |  8.8/15           |  8.5/15           |
> | IOM                            |  8.7/15           |  7.5/15           |
> | ICT **(ours)**                 |  2.8/15           |  1.5/15           |
>
>  > Limited technical novelty
>
> Thank you for your observations on technical novelty. We acknowledge our work's basis in co-teaching and sample reweighing, but emphasize that the novelty lies in adapting these methods to the unique context of offline MBO. This includes:
>
> 1. Management of Noise: Unlike traditional applications focused on label noise in existing datasets, our method employs a pseudo-labeler to dynamically generate data around the current optimization point. This new approach enables us to utilize co-teaching and sample reweighing to enhance proxy robustness along the gradient optimization path, different from their typical use.
>
> 2. Application to Regression vs Classification: Although co-teaching and sample reweighing often target classification tasks, our work creatively leverages these techniques for a regression task within offline MBO by leveraging the task characteristics.
>
> 3. Use of the Offline Dataset: In the domain of meta-learning-based sample reweighing, previous work often necessitates a clean validation set for updating sample weights. Our work instead utilizes an offline dataset for this purpose, further demonstrating novelty in our adaptation.
>
> Our work's novelty, thus, lies in the inventive adaptation of existing techniques to new challenges, enhancing offline MBO performance. The pseudo-labeler's role in this process underscores our method's unique contributions.
>
>  > Presentation can be polished
>
> Thank you for your constructive feedback on the presentation of our paper.
>
> We recognize the redundancies and will work to eliminate them in the abstract, introduction, method, and conclusion.
>
> We appreciate your suggestions regarding Figures 1 and 2. While we believe that the necessary information for understanding these figures is provided in close proximity in the text, we understand that the captions themselves could be more informative. While the captions were concise to avoid repetition, we recognize they could be more informative. Recognizing the importance of self-contained captions, we will revise them to be more comprehensive in the final version of our paper.
>
> ## Overall
> Does this address your concerns? We value your feedback and look forward to further discussion. Thank you.
>
>     [1] Mackenzie, Brandon, et al. "Design-Bench: Benchmarks for Data-Driven Offline Model-Based Optimization." arXiv preprint arXiv:2103.08738 (2021).

---

> > ### Comment · Reviewer_P8Ru · 2023-08-13
> > **Concerns addressed**
> >
> > Thank you for the rebuttal. After reading other reviews and responses, I have increased my rating to acceptance.
> >
> > It would be greatly appreciated if the authors could include the responses, such as clarifications on the MBO-specific novelties over the prior co-teaching, in the revised manuscript. This could be accommodated in the additional space created by reducing redundancies.

---

> > > ### Author Response · Authors · 2023-08-13
> > > **Thanks for your feedback**
> > >
> > > Thank you for your feedback and the adjusted score. We'll ensure that clarifications on MBO-specific novelties, along with other responses, are included in the final manuscript, utilizing the space created by reducing redundancies.

---

### Official Review · Reviewer_j29o · 2023-07-23

**Soundness:** 3 good
**Presentation:** 3 good
**Contribution:** 3 good
**Rating:** 6
**Confidence:** 4

**Summary:**

The work proposes an Importance-aware Co-Teaching for Offline Model-based Optimization (ICT) as a solution for Offline model-based optimization problems. The ICT algorithm is comprised of 2 steps: (i) pseudo-label-driven co-teaching and (ii) per-sample reweighting.

Overall the approach aims to learn an ensemble of methods where the targets predicted by one is treated as the pseudo-label for the other models. Furthermore, the more confident samples with low losses are used to give feedback to the proxy model.

**Strengths:**

* The paper proposes their intuition backed by two clearly laid out steps to walk the readers through the working of the model.
* The extensive results on multiple datasets of both continuous and discrete domains help better bolster their approach.
* The ablation of each step is useful to better weigh each design choice of ICT

**Weaknesses:**

* Some of the results in Table 1, Table 2 are within standard deviation. I would advice the authors to also perform a p-test to better strengthen their claims.
* It would be nice to expand more on the per-sample weighting part. My knowledge on this subject often uses a validation data to as a proxy for minimizing loss, similar to Meta-weight-Net [1]. In the current training regime, it is not clear to me why the model cannot converge to the trivial solution of a zero vector as per-sample weights.

[1] Shu, Jun, et al. "Meta-weight-net: Learning an explicit mapping for sample weighting." Advances in neural information processing systems 32 (2019).

**Questions:**

I am interested in understand two design choices taken in this work:

* In co-teaching, the two models other than proxy share knowledge about top-K points which both have learnt very well. If the K points are in-distribution and all models have learnt them perfectly, I see little to gain from this step. I was thinking using the worst-K points might be more interesting, as I am not sure how the training dynamics would play out there.
* I per-sample weighting, i am curious how the current proposed regime fares compared to other solutions such as RGD [2] which have been shown to achieve improved generalization inspired by DRO literature. RGD would also remove the need to learn another meta-network  However, this again focusses on giving more importance to samples the model is weak at, and roughly connects back to the first point.

[2] Kumar, Ramnath, et al. "Stochastic Re-weighted Gradient Descent via Distributionally Robust Optimization." arXiv preprint arXiv:2306.09222 (2023).

**Limitations:**

The approach is in the domain of optimization, and does not directly have any negative impact by itself. The authors have provided a nice paragraph covering this aspect in their paper.

---

> ### Author Rebuttal · Authors · 2023-08-04
>
> ## General Reply
>
> Thank you for your insightful and constructive feedback! We commit to attentively revising the manuscript based on your remarks, as we describe in detail below.
>
> ## Weakness
>
> >  I would advice the authors to also perform a p-test to better strengthen their claims.
>
> We appreciate your advice to perform a p-test. We have run a statistical significance test (Welch's t-test) between our method and the second-best method. The p-values are 0.437 on SuperC, 0.004 on Ant, 0.009 on D'Kitty, 0.014 on Hopper, 0.000 on TF8, 0.045 on TF10, 0.490 on NAS. This demonstrates our method has statistically significant improvement over other methods in a majority of tasks (5/7), further corroborating the assertions made in our manuscript. We will add the following sentence in Line 257: " We have further run a Welch's t-test between our method and the second-best method, obtaining p-values of 0.437 on SuperC, 0.004 on Ant, 0.009 on D'Kitty, 0.014 on Hopper, 0.000 on TF8, 0.045 on TF10, 0.490 on NAS. This demonstrates statistically significant improvement in 5 out of 7 tasks, reaffirming the effectiveness of our method."
>
> > It would be nice to expand more on the per-sample weighting part. In the current training regime, it is not clear to me why the model cannot converge to the trivial solution of a zero vector as per-sample weights.
>
> Thank you for your feedback. As you noted, methods such as Meta-weight-Net [1] leverage validation data to guide the optimization of sample weights within the training set. Similarly, our approach employs the offline dataset to optimize the sample weights of the pseudo-labeled data. Here the offline dataset plays a similar role as the validation data.
>
> To address your concern about the potential for convergence to a trivial solution of zero for the per-sample weights, we would like to clarify that the weights aren't being updated through the direct optimization of the weighted sample loss in Eq. (5):
> $$
> \boldsymbol{\theta}^{*}(\boldsymbol{\omega}) = \arg\min_{\boldsymbol{\theta}}\frac{1}{K} \sum_{i=1}^{K} \boldsymbol{\omega_i} (f_{\boldsymbol{\theta}}(\boldsymbol{x}^s_i) - \bar{y}^s_i)^2.
> $$
> Instead, our approach is to update the sample weights via the loss of the offline dataset in Eq. (7), a process that is detailed in lines 170 to 181 of our manuscript:
>
> $$
> \mathcal{L}(\boldsymbol{\theta}^*(\boldsymbol{\omega}))= \arg\min_{\boldsymbol{\omega}}\frac{1}{N}\sum_{i=1}^N (f_{\boldsymbol{\theta}^*(\boldsymbol{\omega})}(\boldsymbol{x}_i) - y_i)^2.
> $$
>
> In line 180, we highlight that our meta-learning optimization strategy is designed to allocate higher weights to samples that exhibit similar gradients to those found in the offline dataset. This methodology helps us to avoid convergence to trivial solutions, thus enhancing the robustness and effectiveness of our approach.
>
>
> ## Questions
> > In co-teaching, the two models other than proxy share knowledge about top-K points which both have learnt very well.
>
> Thank you for your interesting observation. There are three key points to address your comment:
>
> 1. The top-K points are more likely to be out-of-distribution as they are sampled along the gradient optimization path, aiming for higher-scoring designs than the offline dataset.
>
> 2. Your point about the top-K points being well learned by all proxies, if they are in-distribution, is insightful. However, it's important to consider that our proxies are initialized randomly and, even though they are pretrained on the same offline dataset, their learning trajectories  differ. Consequently, each proxy grasps different knowledge, providing complementary perspectives, much like ensemble learning.
>
> 3. As for focusing on the worst-K points, these typically correspond to samples where the two proxies largely disagree. Generally, these high-loss samples contain misleading information and directing the third proxy to fine-tune based on these samples could degrade its performance as we demonstrate on TF8 and Dkitty benchmarks in the table below.
>
> | Method  | Best-K Loss (Ours) | Worst-K Loss   |
> |---------|--------------------|----------------|
> | TF8     | **0.958 ± 0.008**  | 0.861 ± 0.058  |
> | D'Kitty | **0.968 ± 0.020**  | 0.958 ± 0.009  |
>
> Thus, we believe our current approach of sharing knowledge about the top-K points is justified and leads to more robust learning dynamics.
>
>
> > I per-sample weighting
>
> Thank you for suggesting the RGD method as a comparable solution in per-sample weighting. Though this work was published after our submission deadline, we have conducted additional experiments on the TFB8 and Dkitty benchmarks:
>
> | Method  | Sample Reweighting (Ours) | RGD-EXP        |
> |---------|---------------------------|----------------|
> | TF8     | **0.958 ± 0.008**         | 0.906 ± 0.058  |
> | D'Kitty | **0.968 ± 0.020**         | 0.955 ± 0.014  |
>
> Although RGD is efficient and theoretically effective, our results were slightly better, possibly due to our method's ability to utilize information from the static dataset.
>
> To address this important work and offer a comprehensive perspective, we will update our manuscript to include a discussion of RGD. At line 313, we will add: "Inspired by distributionally robust optimization, recent work [2] proposes a re-weighted gradient descent algorithm that provides an efficient and effective means of reweighting." Additionally, we will include the comparative experimental results in the Appendix and add a reference at Line 271: "Our reweighting module is also compared with the recently proposed RGD method [2] as detailed in the Appendix."
>
>     [1] Shu, Jun, et al. "Meta-weight-net: Learning an explicit mapping for sample weighting." Advances in neural information processing systems 32 (2019).
>     [2] Kumar, Ramnath, et al. "Stochastic Re-weighted Gradient Descent via Distributionally Robust Optimization." arXiv preprint arXiv:2306.09222 (2023).

---

> > ### Comment · Reviewer_j29o · 2023-08-11
> >
> > Thank you for your detailed response. I have read through most of the comments by other reviewers, as well as the rebuttal.
> > This clears most of my questions and concerns. Thus, I have increased my score.

---

> > > ### Author Response · Authors · 2023-08-11
> > > **Thank you very much for your prompt feedback**
> > >
> > > Thank you for your prompt feedback. We appreciate your review of our responses and others' comments. It's encouraging to see that our clarifications resolved your concerns, and we will make the necessary revisions to the paper as discussed.

---

### Decision · Program_Chairs · 2023-09-21

**Decision:**

Accept (poster)

**Comment:**

The paper introduces an Importance-aware Co-Teaching (ICT) approach for offline model-based optimization to tackle the out-of-distribution challenge. All reviewers find the idea of utilizing a pseudo-labeler to generate training data interesting and valuable. The effectiveness of the method is demonstrated through ablation studies conducted on both continuous and discrete tasks. The proposed solution for handling inaccuracies in pseudo-labeling using a meta-learning framework is reasonable. The paper is praised for its lucidity, significance in addressing out-of-distribution issues, and sufficient experimental validation. Overall, the meta-reviewer deems this paper's contribution worthy of publication. The authors are encouraged to incorporate the reviewers' feedback while preparing the final version.